# Design of Anti-Tumor RNA Nanoparticles and Their Inhibitory Effect on Hep3B Liver Cancer

**DOI:** 10.3390/biom16010045

**Published:** 2025-12-26

**Authors:** Shuyi Sun, Ling Yan, Zhekai Liu, Weibo Jin

**Affiliations:** 1Key Laboratory of Plant Secondary Metabolism and Regulation of Zhejiang Province, College of Life Sciences and Medicine, Zhejiang Sci-Tech University, Hangzhou 310018, China; 2023332864117@mails.zstu.edu.cn (S.S.); 202130802168@mails.zstu.edu.cn (L.Y.); zl6713@nyu.edu (Z.L.); 2Zhejiang Sci-Tech University Shaoxing Academy of Biomedicine, Shaoxing 312366, China

**Keywords:** RNAi therapy, liver cancer, RNA nanoparticles, *hTERT*, *FGFR1*, *BIRC5*

## Abstract

RNA interference (RNAi) holds promise as a gene-silencing therapy for liver cancer but faces challenges related to siRNA instability, short half-life, and inefficient cellular uptake. In this study, we designed a self-assembling RNA nanoparticle targeting three oncogenes—*hTERT*, *BIRC5*, and *FGFR1*—key drivers of cancer progression. These RNA nanoparticles demonstrated enhanced stability and specificity, eliminating the need for conventional toxic delivery carriers. Functional assays revealed that the nanoparticles effectively suppressed the proliferation, migration, tumor growth and apoptosis of a Hepatocellular carcinoma cell line, Hep3B. The nanoparticles exhibited excellent safety and efficacy in xenograft model mice, without off-target toxicity. This work introduces a scalable, biocompatible RNA nanoparticle platform with multi-targeting capability, paving the way for improved RNAi-based therapeutics. Our findings offer a promising strategy for advancing personalized cancer therapies and underscore the broader potential of RNA nanotechnology in addressing complex malignancies.

## 1. Introduction

Cancer remains a significant global health challenge, with 19.3 million new cases and 10 million cancer-related deaths annually [1]. Despite advances in diagnostics and treatments, many cancers, including liver cancer, remain difficult to treat due to their aggressive nature and resistance to conventional therapies [2]. Hepatocellular carcinoma ranks sixth worldwide and is also the second leading cause of cancer-related death [3]. Cancer progression is driven by complex dysregulation of gene expression, with oncogenes activated and tumor suppressors silenced, resulting in unchecked cell proliferation, invasion, and metastasis [4]. Targeting these genetic alterations offers a promising avenue for more precise therapies.

RNA interference (RNAi) has emerged as a promising approach in cancer therapeutics, functioning as a precision tool to silence genes implicated in tumorigenesis [5]. Compared to traditional therapies, RNAi provides superior specificity and sensitivity by targeting genes post-transcriptionally, preventing the production of oncogenic proteins [5,6]. This is particularly advantageous for genes that are difficult to inhibit with traditional drugs, allowing for a more targeted therapeutic approach. RNAi also enables reversible gene silencing without altering the genome permanently, potentially minimizing off-target effects [7]. Given these advantages, RNAi has been extensively researched for modulating gene expression in cancer, demonstrating particular promise in treatment-resistant malignancies [8].

Despite these benefits, the clinical application of RNAi has been limited by challenges in stability [9], delivery [10], and effective targeting [11] in vivo. Small interfering RNAs (siRNAs), the core mediators in RNAi, are vulnerable to rapid degradation in the bloodstream, limited in their ability to cross cell membranes, and generally exhibit a short half-life [12,13]. Specialized delivery systems have therefore been essential for effective RNAi therapy, as siRNAs alone cannot reach therapeutic concentrations in target cells [14]. This underscores the need for innovative delivery systems that improve siRNA stability and cellular uptake to maximize the therapeutic potential of RNAi in cancer treatment.

Conventional approaches to RNAi delivery have predominantly relied on nanocarriers, such as cationic liposomes and polymer-based systems [15], which help protect siRNAs from degradation and enhance cellular uptake through electrostatic interactions with the cell membrane [16]. A further example of NP-mediated siRNA delivery comes from Meng et al.’s study [17], in which siRNA loaded into PLGA-based NP was used to specifically silence *circROBO1*. This circRNA is involved in hepatocellular carcinoma progression controlling the *miR-130a-5p*/CCNT2 axis. However, these carriers often induce side effects, including cytotoxicity [18], immunogenicity [19], and non-specific uptake [20], which compromise therapeutic precision and safety.

RNA nanoparticles (RNA NPs) have emerged as a promising alternative to these traditional carriers. These nanoparticles, which are capable of self-assembling into stable structures, offer a unique advantage in delivering RNAi agents directly to target cells without the need for toxic delivery vehicles. RNA NPs can enhance the stability and specificity of siRNAs, while reducing the risk of off-target effects [21]. Despite their potential, RNA NPs face challenges such as high production costs, complex synthesis, and difficulties in achieving reproducible self-assembly in therapeutic settings [22]. Based on current research on liver cancer and siRNA therapy, and aligned with our lab’s focus, we present an innovative approach to RNAi therapy for liver cancer by designing RNA nanoparticles that self-assemble to target three key oncogenes—*hTERT*, *BIRC5*, and *FGFR1*—each critical to cancer cell proliferation, survival, and metastasis [23,24,25]. We investigate the siRNA’s effects on proliferation, migration, and apoptosis in Hep3B cells in vitro and assess its impact on target gene mRNA and protein expression. Furthermore, we validate its inhibitory effect on tumor growth in a mouse xenograft model, offering a novel strategy for liver cancer treatment. In addition, our approach eliminates the need for conventional delivery systems, enhancing the stability and specificity of siRNA while minimizing potential off-target effects and toxicity. By targeting multiple critical oncogenes, we aim to offer a comprehensive therapeutic solution that improves the precision and efficacy of RNAi-based treatments for cancer.

## 2. Materials and Methods

### 2.1. Ethics Statement (IRBS)

This experiment complies with national laboratory animal welfare ethics guidelines, and all procedures were approved by the Animal Experiment Ethics Committee of Zhejiang Sci-tech University. SPF-grade mice were used and housed in a barrier environment meeting GB14925 standards [26], with free access to food and water. Pain and stress were strictly controlled. Human endpoints were applied post-experiment. This study adhered to the 3R principles to minimize animal usage.

### 2.2. Cell Lines and Cell Culture

Human Hep3B liver cancer cells and Vero cells were kindly provided by Prof. Yigang Wang, Zhejiang Sci-tech University (Hangzhou, China). The Vero cell line was kindly provided by Prof. Yulong He (Hangzhou, Zhejiang Sci-tech University). Both cell lines were cultured in Dulbecco’s Modified Eagle’s Medium (DMEM; Cytiva, Marlborough, MA, USA) supplemented with 10% fetal bovine serum (FBS) (FBS; Cegrogen, South America, Germany) and 1% penicillin–streptomycin (Beyotime, Shanghai, China). Cells were maintained at 37 °C with 5% CO_2_ in a humidified incubator.

### 2.3. Experimental Animals

The mice used in the animal experiment were 4-week-old female BALB/c nude mice purchased from Hangzhou Hangsi Biotechnology Co., Ltd. (Hangzhou, China) (production license SCXK (Zhejiang) 2022-0005), which weighed from nine to fifteen grams. The nude mice were kept in a clean animal room with a temperature of 22 ± 2 °C, a relative humidity of around 70%, and a light and dark cycle of 12 h each. Five mice were housed per cage with wood shavings as beds, which were changed regularly. Additionally, we provided autoclaved water and standard feed. All animal experiments were conducted in accordance with animal protection guidelines and approved by the Experimental Animal Welfare Ethics Committee of the College of Life Sciences, Zhejiang Sci-tech University (protocol code 20220304-16, approved date: 4 March 2022).

### 2.4. Design, Synthesis, and Extraction of Nanoparticles

Using siDirect 2.0 program, multiple suitable siRNA target sites were identified on the three oncogenes of *hTERT*, *BIRC5* and *FGFR1*. Based on siRNA’s molecular characteristics and mechanism of action, we chose siRNA molecules with strong specificity and high RNAi efficiency. By concatenating the above-mentioned siRNA using RNA structural elements, a long chain RNA molecule was obtained, which was the RNA nanoparticle used in this experimental study, named F1B3hT1-RNA. The non-targeted RNA nanoparticle used in subsequent experiments was named NC-RNA. The coding sequences of the synthesized RNA were sent to GenScript Biotech Corp. (Nanjing, China) for synthesis and inserted between the Bgl II and Bpu I sites of the PET28a vector to obtain a positive plasmid.

Positive plasmids were transformed into competent *E. coli* cells, and single colonies were selected and cultured overnight in LB liquid medium at 37 °C and 220 rpm. The cultured cells were further expanded at a ratio of 1:100 until the optical density (OD) value was 0.5–0.6. IPTG solution with a mother liquor concentration of 100 mg/mL was added at a ratio of 1:1000 for induction, and the cells were induced for more than 4 h on a shaking table at 37 °C and 220 rpm. Then, the induced cells were centrifuged at 8000 rpm and 4 °C for 5 min, and the supernatant was discarded. The precipitate was added to the corresponding volume of extraction solution (Tris HCl and magnesium acetate solution volume ratio of 1:100) at a ratio of 1:10 of the bacterial volume. Bacterial bodies were resuspended, mixed with RNA extraction phenol reagent with the same volume, and shaken vigorously. The bacterial cells were thoroughly mixed with the reagent, incubated in a water bath at 64 °C for 20 min, and then immersed in an ice bath for 5 min. Chloroform of the same volume as the extraction solution was added; the mixture was shaken vigorously and centrifuged at 12,000 rpm at 4 °C for 20 min, and then the supernatant was transferred into a new centrifuge tube. More than 1/2 volume of chloroform was added to the supernatant again, and the mixture was vigorously shaken and centrifuged at 12,000 rpm at 4 °C for 15 min; the supernatant was then transferred into a new centrifuge tube. The supernatant was the crude extract of RNA nanoparticles. A 7.5 M LiCl solution at double the supernatant volume was added and left to settle overnight at −20 °C, after which it was centrifuged at 12,000 rpm at 4 °C and left to settle for 20 min; then the precipitate was collected. Finally, 5 µL of RNA solution was taken, and horizontal gel electrophoresis (80 V, 30 min) was conducted in 2% agar gel to detect the extraction of RNA nanoparticles.

### 2.5. MTT Assay for Cell Viability

MTT assay was used to detect the activity of RNA nanoparticles on Vero cell growth. The cells were inoculated into a 96-well plate at a concentration of 1 × 10^4^ cells/mL, with a volume of 100 µL per well of culture medium. Then, the 96-well plate was cultured in a 37 °C, 5% CO_2_ incubator for 12 h. After the cells adhered to the wall, we added RNA nanoparticles at concentrations of 50 ng/µL, 100 ng/µL, 200 ng/µL, 500 ng/µL, 800 ng/µL, and 1000 ng/µL to the culture wells separately, with 3 replicate wells set for each concentration. After 48 h of cultivation, we discarded the culture medium and added 20 µL of MTT solution (5 mg/mL, prepared in PBS, Solarbio, Beijing, China) to each well. We continued incubating the cells for 4 h before terminating the culture. The supernatant was carefully aspirated from each well, and 150 µL of DMSO (Solarbio, Beijing, China) reagent was added. The mixture was shaken rapidly on a horizontal shaker for 10 min to fully dissolve the crystals. The optical density (OD) value at a wavelength of 490 nm was detected using the Microplate Reader (FLUO star Omega, Offenburg, Germany). The relative cell viability percentage of each experimental group was calculated based on the control group, and the half-maximal inhibitory concentration (IC_50_) value of drug concentration was plotted, with RNA nanoparticle concentration as the horizontal axis and cell viability as the vertical axis.

### 2.6. MTT Assay for Cell Proliferation

MTT assay was used to detect the inhibitory effect of RNA nanoparticles on the growth of Hep3B liver cancer cells. The cells were inoculated into a 96-well plate at a concentration of 1 × 10^4^ cells/mL, with a volume of 100 µL per well of culture medium. After plating, the 96-well plate was cultured in a 37 °C, 5% CO_2_ incubator for 12 h. Based on the half maximal inhibitory concentration (IC_50_) value obtained from Vero cells in the previous stage, we selected an RNA nanoparticle concentration of 150 ng/µL for subsequent experiments, added the corresponding volumes of 150 ng/µL RNA concentration to the culture wells, and took out a 96-well plate for detection at 24 h, 48 h, 72 h, and 96 h culture times. Similarly, 20 µL of MTT solution was added to each well. After incubation in the incubator for 4 h, 150 µL of DMSO solution was added and shaken for 10 min in order to fully dissolve the crystals. The optical density (OD) value at a wavelength of 490 nm was detected using the Microplate Reader, and the cell proliferation curve was plotted.

### 2.7. Cell Migration Ability Detection

Using scratch assay to detect the effect of RNA nanoparticles on the migration ability of Hep3B cells. Inoculated cells into a 6-well plate at a concentration of 5 × 10^5^ cells/mL, with a volume of 2 mL per well of culture medium. After overnight cultivation, the cell fusion rate reached 100%. A 200 µL pipette tip was used to scratch the cells perpendicular to the horizontal line behind the 6-well plate. The upper layer of culture medium was discarded, and the cells were washed three times with PBS buffer. In the experimental group, RNA nanoparticles at a concentration of 150 ng/µL were added to the 6-well plate, with three replicate wells set for each treatment. The 6-well plate was placed under an inverted microscope, and pictures of the cell scratch situation in each group were taken at 0 h. The cell gap width at 0 h was used as a control for later data processing. The 6-well plate continued to be cultured in a 37 °C, 5% CO_2_ incubator. When the drug treatment time was reached, such as 24 h, 48 h, and 72 h, we continued to take photos under the microscope and recorded the cell migration at different times. Analyzing scratch images of cells from different time periods and groups can detect the migration ability and scratch repair ability of each group of cells. The calculation formula for cell migration rate was cell migration rate (%) = (scratch distance at 0 h − 24 h/48 h/72 h)/scratch distance at 0 h × 100%.

### 2.8. Cell Apoptosis Detection

Annexin V-FITC/PI was used to detect the effect of RNA nanoparticles on Hep3B cell apoptosis. Cells at a concentration of 5 × 10^5^ cells/mL were inoculated into a 6-well plate, with a volume of 2 mL of culture medium per well. The 6-well plate was cultured in a 37 °C, 5% CO_2_ incubator for 12 h. After the cells adhered to the wall, we added the corresponding volume of 150 ng/µL RNA concentration to the culture wells. A 6-well plate was taken out for experiments at 24 h, 48 h, and 72 h. The specific operation should be carried out according to the instructions of the cell apoptosis detection kit (A211, Vazyme, Nanjing, China). Flow cytometry was used for sample analysis at an excitation wavelength of 488 nm. FITC’s green fluorescence was detected in the FL1 channel, and PI’s red fluorescence was detected in the FL2 channel. FlowJo10 software (BD, Franklin Lakes, NJ, USA) was used to draw a cross gate for data analysis. According to the designated cross gate, cells could be divided into four parts: live cell population, early apoptotic cells, late apoptotic cells, and cell death caused by mechanical damage. By comparing with control cells, the early and late apoptotic cells in the experimental group can be detected.

### 2.9. Detection of Target Gene Expression

RT-qPCR was used to detect the expression of target genes in Hep3B cells by RNA nanoparticles. The cells were inoculated into a 6-well plate at a concentration of 5 × 10^5^ cells/mL, with a volume of 2 mL per well of culture medium. The 6-well plate was cultured in a 37 °C, 5% CO_2_ incubator for 12 h. After the cells adhered to the wall, we added the corresponding volumes of 150 ng/µL RNA concentration to the culture wells. The cells were collected at 24 h, 48 h, and 72 h of culture time. The supernatant was discarded, and the cells were washed twice with PBS buffer. Then, 500 µL of “Buffer RL” was added to each well of cells for digestion and lysis. The instructions of the Cell Total RNA Extraction Kit (RC101-01, Vazyme, Nanjing, China) were followed to perform reverse transcription according to the instructions of the HiFiScript cDNA Synthesis Kit (Kangwei, Taizhou, China). The various reagents required for the experiment were melted on ice in advance, and the reverse transcription produced after the completion of each reaction could be directly used for fluorescence quantitative PCR reaction or stored at −20 °C for a long time. The SYBR qPCR Master Mix reaction reagent was provided by Novozymes, and the cDNA product obtained from the above experiment was subjected to qPCR amplification. In 8 tubes, 20 µL of reaction system was added to each well in the order of sample loading. The qPCR cycle reaction conditions began with a pre-denaturation step (95 °C, 30 s) and then proceeded through 40 cycles of denaturation (95 °C, 3–10 s) and combined annealing/extension (60 °C, 10–30 s) with fluorescence acquisition. Then, it ended with a melting curve (95 °C, 15 s; 60 °C, 20 s; 95 °C, 15 s). Flod changes were calculated employing the 2^−ΔΔCt^ method, where ΔΔCt = (Ct, target − Ct, inner) _treatment_ − (Ct, target − Ct, inner) _control_ [27].

### 2.10. Detection of Target Protein Expression

Western Blot was used to detect the expression of target proteins in Hep3B cells by RNA nanoparticles. Cells at a concentration of 5 × 10^5^ cells/mL were inoculated into a 6-well plate, with a culture medium volume of 2 mL per well. The 6-well plate was cultured in a 37 °C, 5% CO_2_ incubator for 12 h. After the cells adhered to the wall, we added the corresponding volumes of 150 ng/µL RNA concentration to the culture wells. And cell collection was performed at 48 h, 72 h, and 96 h culture times. The supernatant was collected in a 5 mL EP tube and washed twice with PBS buffer, and 150 µL of Loading Buffer containing lysis buffer was added to each well. The lysis buffer was collected in a 5 mL EP tube and incubated on ice for 30 min to fully lyse the cells. The sample could be stored at −80 °C. The electrophoresis conditions for concentrated gel were 80 V and 30 min. When the sample was pressed into a line in the concentrated gel, we adjusted the voltage of the separation gel to 120 V and 60 min until bromophenol blue reached the bottom. Subsequently, the membrane was transferred and sealed overnight at 4 °C with 5% skim milk powder (2.5 g skim milk powder + 1 × TBST buffer 50 mL). The membrane was washed three times with 1 × TBST buffer, and the corresponding GAPDH (1:5000, HUABIO, Hangzhou, China), hTERT (1:1000, HUABIO, China), BIRC5 (1:1000, Proteintech, Wuhan, China), and FGFR1 (1:1000, Proteintech, Wuhan, China) were diluted with 1 × TBST buffer. The PVDF membrane was placed in the diluted primary antibody solution and incubated on a shaker at 70 rpm for 2 h at room temperature. After the completion of the primary antibody incubation, the membrane was washed three times for 10 min each time. The secondary antibody (1:50,000, HUABIO, Hangzhou, China) was diluted with 1 × TBST buffer, and the membrane was placed in the diluted secondary antibody solution. The membrane was incubated at room temperature for 2 h on a shaker at 70 rpm. After the secondary antibody incubation was completed, the membrane was washed 3 times, and 500 µL of developer was evenly dropped onto each membrane. The protein bands were imaged using a chemiluminescence system to analyze the expression of the target protein at different experimental times.

### 2.11. Establishment of Subcutaneous Xenograft Tumor Model

Four-week-old BALB/c female mice were normally housed in the animal room for one week. Each nude mouse was inoculated with 100 µL of Hep3B cell suspension diluted with PBS buffer in the subcutaneous tissue of the right back, with a dose of 5 × 10^6^/mL Hep3B cell suspension per mouse. When the tumor volume of the mice reached 50–80 mm^3^, the model mice were randomly divided into four groups: tail vein injection of extract (100 µL) as the control group, tail vein injection of 2.5 mg/kg nano RNA (100 µL) group, tail vein injection of 5 mg/kg nano RNA (100 µL) group, and intratumoral injection of 30 mg/kg nano RNA (100 µL) group. The tail vein injection group was injected every 3 days, with a total of 6 injections during the experiment. The intratumoral injection group was injected again on the 8th day, with a total of 2 injections during the experiment. Starting from the day of RNA injection, mice were weighed every two days, and the length (a) and length (b) of the tumor were measured using a vernier caliper. The calculation formula for tumor volume was V = ab^2^/2. The tumor mice were continuously treated with RNA drugs. When the maximum tumor volume of the control group mice reached 2000 mm^3^, all experimental mice were euthanized by removing their necks and cryopreserved before being subjected to harmless disposal [28], and the tumor volume size of each group was recorded. Dissected the mice, isolated subcutaneous tumors from each mouse, weighed the tumor weight, and fixed the mouse liver and kidney tissues in 4% paraformaldehyde before sending them to Hangzhou Haoke Biotechnology Co., Ltd. (Hangzhou, China) for HE staining analysis.

### 2.12. Statistical Analysis

The experimental data analysis and chart creation were performed using GraphPad Prism9 software (GraphPad, San Diego, CA, USA), and quantitative data was presented in the form of means ± standard deviation (means ± SD) or means ± standard error of the mean (means ± SEM). IC_50_ determination was performed using non-linear regression analysis (log [concentration] vs. Control). Single-factor, multiple-sample analysis uses one-way ANOVA and multiple comparisons to compare the differences between multiple groups of data. Two-factor, multiple-sample analysis uses two-way ANOVA and multiple comparisons to compare the differences between multiple groups of data. When *p* < 0.05, it can be considered that there are significant differences between different groups.

## 3. Results

### 3.1. Expression and Structural Characterization of F1B3hT1-RNA Nanoparticles in Escherichia coli

To evaluate the feasibility of using RNA nanoparticles for liver cancer therapy, we first designed and synthesized the F1B3hT1-RNA nanoparticles, which target three critical oncogenes—*hTERT*, *BIRC5*, and *FGFR1*. Six siRNA molecules targeting these genes were selected for their strong specificity and high RNA interference efficiency (Appendix A). After concatenating the siRNAs using RNA structural elements, a long-chain RNA molecule that could form a specific nanostructure was obtained and named F1B3hT1-RNA (Figure 1a). The nanoparticle’s structure was confirmed using secondary predictions through the RNAfold program (Figure 1b). Furthermore, the RNA nanoparticles were successfully expressed in *Escherichia coli*, which could greatly reduce the high production cost of RNA nanoparticles (Figure 1c). The tertiary structure of F1B3hT1-RNA was observed by cryo-electron microscopy (Figure 1d). After dynamic light scattering (DLS), particle size distribution was concentrated in the 70–90 nm range with a narrow distribution, indicating uniform size and good dispersion (Figure 1e, Appendix A).

### 3.2. The Effect of F1B3hT1-RNA Nanoparticles on Vero Cell Viability

Cytotoxicity of the F1B3hT1-RNA nanoparticles was assessed in Vero cells to determine the safe dosage range for further experiments. MTT assay revealed that the half inhibitory concentration (IC_50_) of RNA nanoparticles on Vero cells after 48 h of treatment was 499.5 ng/µL (Figure 2, Appendix A). As the concentration of RNA nanoparticles increased, cell viability decreased, with the highest concentration (1000 ng/µL) causing near-complete cell death. These findings suggested that the RNA nanoparticles exert a concentration-dependent cytotoxic effect on non-cancerous cells, which was important for optimizing dosage in future applications. In subsequent experiments, an RNA concentration of 150 ng/µL was selected for exploration because the cell viability was still high at concentrations of 200 ng/µL.

### 3.3. Inhibition of Hep3B Cell Proliferation and Migration by F1B3hT1-RNA Nanoparticles

The anti-proliferative effects of the F1B3hT1-RNA nanoparticles were evaluated in Hep3B liver cancer cells. MTT assays revealed a significant reduction in cell proliferation at 72 and 96 h after treatment with 150 ng/µL of F1B3hT1-RNA nanoparticles, compared to the control and non-targeted RNA treatment groups (Figure 3a, Appendix A). In addition, wound healing assay was performed to evaluate the effect of RNA nanoparticles on cell migration. Results showed that the migration rate was reduced to 36.01% in the experimental group at 72 h, compared to 82.81% in the control group (Figure 3b, Appendix A) and 71.48% in the non-targeted RNA group (Figure 3c, Appendix A). These results suggested that F1B3hT1-RNA nanoparticles effectively inhibited the growth and migration of Hep3B cells, potentially preventing tumor metastasis. These findings highlighted the potential of the F1B3hT1-RNA nanoparticles in suppressing liver cancer cell proliferation.

### 3.4. Induction of Hep3B Cell Apoptosis by F1B3hT1-RNA Nanoparticles

Next, we investigated the apoptotic effects of F1B3hT1-RNA nanoparticles in Hep3B cells using flow cytometry with FITC/PI double staining. As shown in Figure 4a,b (Appendix A), the apoptosis rate of Hep3B cells treated with 150 ng/µL F1B3hT1-RNA nanoparticles increased significantly at 48 and 72 h compared to the control group and non-targeted RNA group. At 72 h, the apoptosis rate in the experimental group reached 25.59%, compared to 4.74% in the control group and 4.19% in the non-targeted RNA group. These results confirmed that the RNA nanoparticle induces apoptosis in Hep3B liver cancer cells, contributing to their anti-cancer effects.

### 3.5. Downregulation of Oncogene Expression in Hep3B Cells by F1B3hT1-RNA Nanoparticles

To understand the molecular mechanisms underlying the therapeutic effects of RNA nanoparticles, we investigated the expression levels of target oncogenes—*hTERT*, *BIRC5*, and *FGFR1*—by quantitative real-time PCR (qRT-PCR). The results showed that treatment with F1B3hT1-RNA nanoparticles led to a significant reduction in the mRNA levels of these oncogenes at 24, 48, and 72 h (Figure 5a–c, Appendix A). Specifically, the mRNA expression of *hTERT* and *BIRC5* was significantly downregulated at all time points, while *FGFR1* expression was markedly reduced at 72 h. In addition, Western blot analysis was performed on Hep3B cells treated with F1B3hT1-RNA nanoparticles to investigate the downregulation of oncogenes at the protein level. Results showed that the protein levels of all three genes were significantly reduced in cells treated with F1B3hT1-RNA nanoparticles at 48, 72, and 96 h (Figure 5d–f, Appendix A). These findings proved that F1B3hT1-RNA effectively inhibited the expression of the target oncogenes at both the mRNA and protein levels, supporting their potential as a therapeutic agent for liver cancer.

### 3.6. In Vivo Tumor Growth Inhibition by RNA Nanoparticles

To evaluate the therapeutic efficacy of F1B3hT1-RNA nanoparticles in vivo, we established a subcutaneous Hep3B xenograft model in nude mice. Tumor-bearing mice were treated with either tail vein injections or intratumoral injections of F1B3hT1-RNA nanoparticles. The results showed that both the 5 mg/kg tail vein injection and 30 mg/kg intratumoral injection significantly reduced tumor growth compared to the control group (Figure 6b, Appendix A). Tumor volume was monitored every two days, and the results showed that both tail vein and intratumoral RNA nanoparticle treatments led to significant tumor growth inhibition compared to the control group (Figure 6c, Appendix A). At the end of the treatment period, tumors from the RNA nanoparticle-treated groups were significantly smaller than those from the control group (Figure 6a). These findings demonstrated that F1B3hT1-RNA nanoparticles effectively inhibit tumor growth in vivo, with favorable therapeutic outcomes.

### 3.7. HE Staining Results of Liver and Kidney in Tumor Mice

To evaluate the potential toxicity of RNA nanoparticles, we performed histological analysis of liver and kidney tissues from treated mice. Hematoxylin and eosin (HE) staining revealed no significant damage to the liver or kidney tissues in the F1B3hT1-treated groups (Figure 7). The liver and kidney tissues showed normal morphology, with no signs of hepatocyte ballooning, inflammation, or necrosis. These results indicated that the RNA nanoparticles did not induce significant toxicity in vital organs.

## 4. Discussion

Recent studies have demonstrated that nanoparticles modified with GALA peptide can achieve efficient gene knockout effect in pulmonary endothelial cells [29]. Building on these findings, this study introduces a novel self-assembling RNA nanoparticle (RNA-NP) system designed to simultaneously silence three key oncogenes—*hTERT*, *BIRC5*, and *FGFR1*. Such a multi-target approach is particularly relevant for malignancies like hepatocellular carcinoma (HCC), where tumor progression and drug resistance often arise from the interplay of multiple dysregulated genes [30,31,32]. Our results show that these RNA-NPs effectively inhibit tumor cell proliferation, migration, and induce apoptosis, offering a promising therapeutic strategy for hepatocellular carcinoma (HCC).

Notably, this carrier-free system circumvents the limitations of traditional delivery carriers. Traditional lipid-based carriers and polymeric nanoparticles though widely used, frequently induce cytotoxicity in vivo [33,34]. RNA nanoparticles, by contrast, exhibit a high degree of biocompatibility due to their natural anionic charge, which reduces the risk of off-target effects and immune activation. Additionally, by using siRNAs as the structural scaffold, the RNA nanoparticles can be engineered to self-assemble into stable structures that provide controlled release and targeted delivery. Their direct biosynthesis in *E. coli* also largely reduces manufacturing costs. Although F1B3hT1-RNA nanoparticle was purified using repeated phenol–chloroform extraction and ethanol precipitation and appeared as a clean single band on agarose gel, quantitative purity analysis and endotoxin measurement will be included in future studies to further improve process standardization and ensure applicability in translational research.

Despite the advantages, several challenges must be addressed for clinical translation. The scalability of RNA nanoparticle synthesis and the potential for immune responses to RNA-based therapeutics are areas that require further investigation. Additionally, the long-term stability and biodistribution of RNA-NPs in vivo need to be thoroughly evaluated to ensure sustained therapeutic effects and minimal off-target interactions. Future studies should focus on optimizing the design of RNA-NPs to enhance their targeting specificity and therapeutic efficacy. Incorporating targeting ligands or utilizing stimuli-responsive materials could further improve the delivery and release of siRNAs at tumor sites. Moreover, exploring combination therapies that integrate RNA-NPs with other treatment modalities, such as chemotherapy or immunotherapy, may offer synergistic effects and address the multifaceted nature of cancer [35]. In addition, the current work is limited by small sample sizes and the absence of comprehensive biodistribution, pharmacokinetic, and immunogenicity analyses. Future studies will address these aspects to determine the suitability of the RNA-NP for further pre-clinical development.

## 5. Conclusions

In this study, we have developed a novel self-assembling RNA nanoparticle system capable of targeting three key oncogenes—*hTERT*, *BIRC5*, and *FGFR1*—that play critical roles in cancer progression. Our findings demonstrate that these RNA nanoparticles effectively suppress cell proliferation and migration, while inducing apoptosis in Hep3B liver cancer cells, offering a promising therapeutic strategy for hepatocellular carcinoma (HCC). Notably, this carrier-free approach eliminates the need for traditional toxic delivery carriers, thereby improving biocompatibility and enhancing specificity. Consistent with the in vitro findings, our in vivo results further confirm the therapeutic efficacy of RNA nanoparticles, showing marked inhibition of tumor growth without causing off-target toxicity. Collectively, these findings support RNA nanoparticles as a safe, efficient, and scalable platform for RNA interference-based liver cancer therapy. Future research should focus on optimizing the delivery system, evaluating long-term effects, and exploring combination therapies to further enhance the therapeutic potential of RNA nanoparticle-based treatments.

## Figures and Tables

**Figure 1 biomolecules-16-00045-f001:**
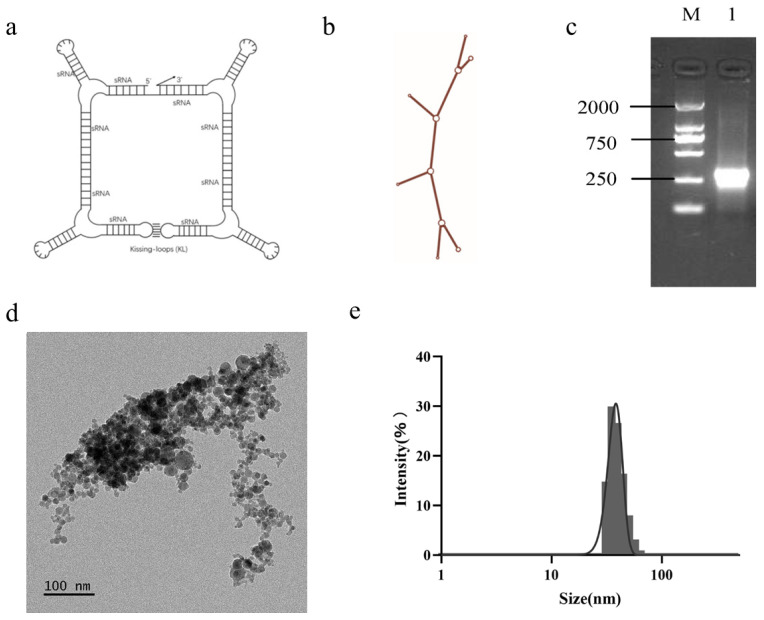
The Structure of RNA Nanoparticles and Their Synthesis in *Escherichia coli*. (**a**) Tertiary structure of F1B3hT1-RNA nanoparticle. (**b**) Secondary structure of F1B3hT1-RNA nanoparticle. (**c**) Expression of F1B3hT1-RNA nanoparticles in *Escherichia coli*. Lane M: DL2,000 DNA Marker; Lane 1: Agarose gel electrophoresis bands of RNA nanoparticles precipitated by LiCl. (**d**) Tertiary structure of F1B3hTI-RNA under scanning electron microscopy. (**e**) Dynamic light scattering (DLS) profile of F1B3hT1-RNA. Original images can be found at Appendix A.

**Figure 2 biomolecules-16-00045-f002:**
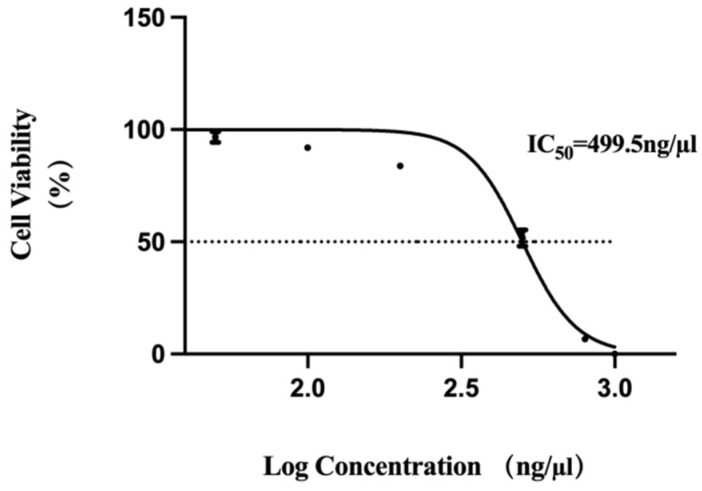
Effect of F1B3hT1-RNA nanoparticle concentration on the growth of Vero cells.

**Figure 3 biomolecules-16-00045-f003:**
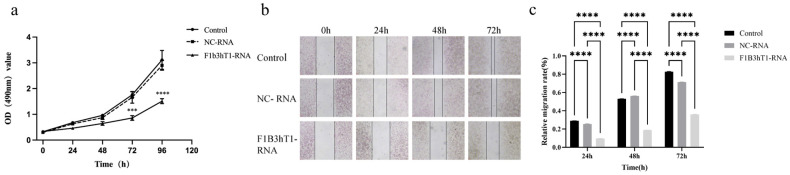
Effect of F1B3hT1-RNA on the proliferation and migration of Hep3B cells. (**a**) The effect of different treatment groups on the proliferation of Hep3B cells. (**b**) The migration of Hep3B cells in different treatment groups at 0 h, 24 h, 48 h, and 72 h. The scale bar was set to 100 pixels. (**c**) Migration rates of Hep3B cells in different treatment groups at 24 h, 48 h, and 72 h. The results are expressed as the means ± SD of three biological replicates. Data was analyzed by two-way ANOVA and multiple comparisons. Asterisks indicate a significant difference ( *** *p* < 0.001, **** *p* < 0.0001) compared to the control.

**Figure 4 biomolecules-16-00045-f004:**
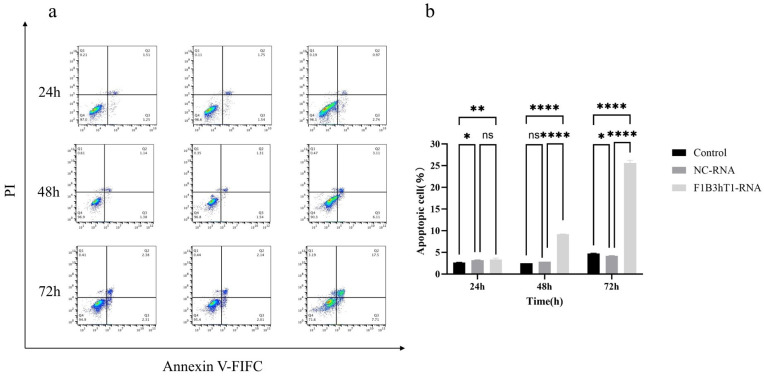
Apoptosis of Hep3b cells induced by F1B3hT1-RNA nanoparticles. (**a**) The apoptosis of Hep3B cells in different treatment groups at 24 h, 48 h, and 72 h. (**b**) Apoptosis rates of Hep3B cells in different treatment groups at 24 h, 48 h, and 72 h. The results are expressed as the means ± SD of three biological replicates. Data was analyzed by two-way ANOVA and multiple comparisons. Asterisks indicate a significant difference (ns > 0.05, * *p* < 0.05, ** *p* < 0.01, **** *p* < 0.0001) compared to the control.

**Figure 5 biomolecules-16-00045-f005:**
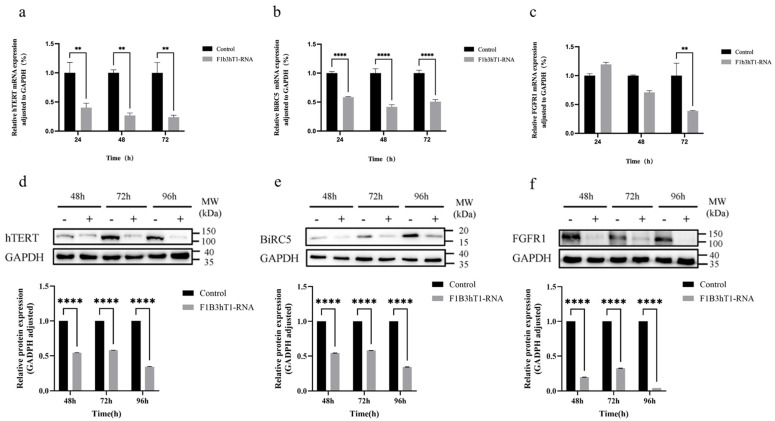
Inhibitory effect of RNA nanoparticles on target gene expression. (**a**–**c**) qRT-PCR was used to detect the mRNA expression levels of three target genes (*hTERT*, *BIRC5* and *FGFR1*) in Hep3B cells at 24, 48, and 72 h in different treatment groups. (**d**–**f**) Western Blot was used to detect the expression levels of target proteins (hTERT, BIRC5 and FGFR1) in different treatment groups at 48 h, 72 h, and 96 h. The results are expressed as the means ± SD of three biological replicates. Asterisks indicate a significant difference (** *p* < 0.01, **** *p* < 0.0001) compared to the control. Original images can be found at Appendix A.

**Figure 6 biomolecules-16-00045-f006:**
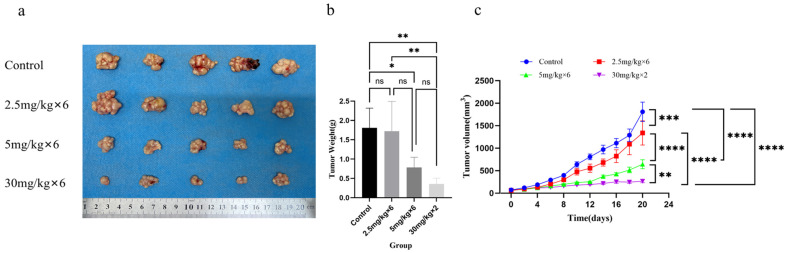
The inhibitory effect of F1B3hT1-RNA on tumor growth in nude mice. (**a**) Anatomical images of tumors in mice treated with RNA nanoparticles in each group. (**b**) Anatomical weighing of tumors in each group of mice. Data was analyzed by one-way ANOVA and multiple comparisons. (**c**) Tumor growth curve of mice during the experiment. The results are expressed as the means ± SEM of three biological replicates. Data was analyzed by two-way ANOVA and multiple comparisons. Asterisks indicate a significant difference (ns > 0.05, * *p* < 0.05, ** *p* < 0.01, *** *p* < 0.001, **** *p* < 0.0001) compared to the control.

**Figure 7 biomolecules-16-00045-f007:**
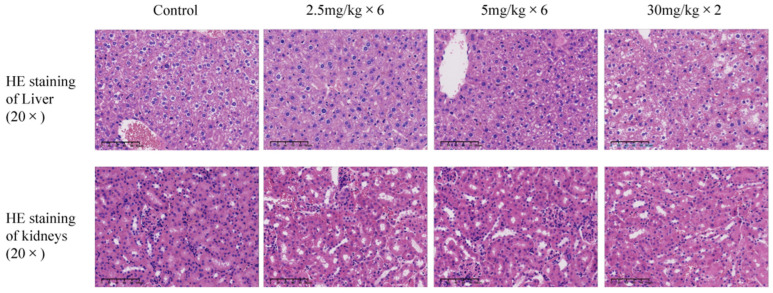
HE staining patterns of liver and kidney in different groups of tumor mice at a magnification of 20×. In the image, the cell nucleus is blue, and the cytoplasm is red.

## Data Availability

The original contributions presented in this study are included in the article/Appendix A. Further inquiries can be directed to the corresponding author.

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
