# Peer review of "Design of Anti-Tumor RNA Nanoparticles and Their Inhibitory Effect on Hep3B Liver Cancer"

_biomolecules, 2025, doi:10.3390/biom16010045_

Round 1
Reviewer 1 Report
Comments and Suggestions for Authors
This manuscript describes the design and biological evaluation of self-assembling RNA nanoparticles that target three oncogenes in hepatocellular carcinoma. The authors report both in vitro and in vivo experiments showing inhibition of Hep3B proliferation, migration, and tumor growth, with limited cytotoxicity in normal cells and no histological evidence of liver or kidney toxicity. The topic is relevant and addresses an important challenge in RNA-based therapeutics. The study is well structured, but the data presentation, statistical rigor, and experimental depth require significant improvement before the manuscript can be considered for publication. Therefore, I warrant major revisions.
Major points
- The manuscript claims novelty for a “self-assembling RNA nanoparticle,” but only in silico folding and agarose gel data are presented. Physical characterization of the nanoparticles (size, morphology, and homogeneity) using DLS and TEM are essential to confirm actual nanostructure formation.
- The dosing rationale and in vivo experimental design need strengthening. The xenograft study uses small groups (n=3), and while error bars and significance are shown, this sample size limits statistical power. Tumor volume data should be plotted over time (with a proper longitudinal model), and body weight curves should be separated from tumor growth data for clarity.
- The reported mean ± SD with asterisks confirms that statistical analyses were performed, but using one-way ANOVA for multiple time points is inadequate. Repeated-measures ANOVA or mixed-effects models should be used, or at least a correction for multiple comparisons applied. Exact P values and replicate numbers should be stated for each figure.
- While the multi-target approach (hTERT, BIRC5, FGFR1) is conceptually interesting, the manuscript lacks evidence that this tri-gene strategy is synergistic or more effective than single-gene targeting. Comparative controls would substantiate the therapeutic advantage.
- The apoptosis, migration, and proliferation results are interesting findings but additional mechanistic assays (e.g., caspase activation, cell-cycle analysis, or Western blot validation of apoptotic markers) would make the conclusions more convincing.
- Safety conclusions are based solely on H&E staining. Serum chemistry (ALT, AST, BUN, creatinine) should be provided to confirm systemic safety and exclude subtle organ toxicity.
- The RNA production and purification workflow described (E. coli expression and phenol–chloroform extraction) may leave residual contaminants. Purity, yield, and endotoxin levels should be reported.
- In discussion section authors should be cautions to claims about scalability and therapeutic translation and more explicitly acknowledge limitations such as small sample sizes, lack of biodistribution data, and untested immunogenicity.
- Figures are of acceptable quality but require improved labeling and consistency. The mismatch between figure captions (e.g., “weight growth curve” vs. tumor volume) needs correction. Include scale bars, replicate counts, and clear legends for all graphs.
Minor point
Define abbreviations at first use, ensure consistent statistical notation, polish grammar and flow (particularly in Methods and Discussion), and harmonize reference formatting.
Author Response
Comments 1: The study is well structured, but the data presentation, statistical rigor, and experimental depth require significant improvement before the manuscript can be considered for publication.
Response 1: Thanks for your kind guidance. We have revised on the manuscript.
Comments 2: The manuscript claims novelty for a “self-assembling RNA nanoparticle,” but only in silico folding and agarose gel data are presented. Physical characterization of the nanoparticles (size, morphology, and homogeneity) using DLS and TEM are essential to confirm actual nanostructure formation.
Response 2: Regarding comments on using DLS and TEM, we completely agree with your concern. We supplement this part of experimental results on “3.1 Expression and structural characterization of F1B3hT1-RNA nanoparticles in E. coli” (Page 7, Line 301-311).
Comments 3: The dosing rationale and in vivo experimental design need strengthening. The xenograft study uses small groups (n=3), and while error bars and significance are shown, this sample size limits statistical power. Tumor volume data should be plotted over time (with a proper longitudinal model), and body weight curves should be separated from tumor growth data for clarity. The reported mean ± SD with asterisks confirms that statistical analyses were performed, but using one-way ANOVA for multiple time points is inadequate. Repeated-measures ANOVA or mixed-effects models should be used, or at least a correction for multiple comparisons applied. Exact P values and replicate numbers should be stated for each figure.
Response 3: Thanks for you highlighting this. We reanalyzed relevant data and you can see on the manuscript.
Comments 4: While the multi-target approach (hTERT, BIRC5, FGFR1) is conceptually interesting, the manuscript lacks evidence that this tri-gene strategy is synergistic or more effective than single-gene targeting. Comparative controls would substantiate the therapeutic advantage.
Response 4: In the article Conversion of Chemical Drugs into Targeting Ligands on RNA Nanoparticles and Assessing Payload Stoichiometry for Optimal Biodistribution in Cancer Treatment, Xu et al. investigated the regulation of methotrexate (MTX) ligand valency on RNA nanoparticles, demonstrating the significance of multivalent targeting in optimizing in vivo biodistribution and therapeutic efficacy. And another study also demonstrated that dual-targeting mRNA nanoparticles achieved significantly higher transfection efficiency in the lungs compared to non-targeting and single-targeting versions, as reported in the article Inhaled mRNA Nanoparticles Dual-Targeting Cancer Cells and Macrophages in the Lung for Effective Transfection.
Comments 5: The apoptosis, migration, and proliferation results are interesting findings but additional mechanistic assays (e.g., caspase activation, cell-cycle analysis, or Western blot validation of apoptotic markers) would make the conclusions more convincing.
Response 5: We sincerely appreciate the reviewer’s insightful suggestion. We fully agree that assays such as caspase activation, cell-cycle profiling, or Western blotting of apoptotic markers would further deepen the mechanistic understanding of how F1B3hT1-RNA nanoparticle regulates apoptosis and proliferation.
At this stage, our study primarily aims to evaluate the therapeutic potential and gene-silencing efficiency of F1B3hT1-RNA nanoparticle targeting hTERT, BIRC5 and FGFR1, and we have provided multiple lines of evidence supporting the antitumor effect, including:
1. Direct gene-silencing data (RT-qPCR and Western blot) confirming that three hTERT, BIRC5 and FGFR1 are significantly down-regulated;
2. Cell viability, migration, and apoptosis phenotypes, which are fully consistent with the downstream biological functions of these three oncogenic targets;
3. In vivo tumor-suppression data, showing that F1B3hT1-RNA nanoparticle effectively inhibits tumor growth.
Given that the primary goal of this study is to establish the efficacy of F1B3hT1-RNA nanoparticle and to validate its multi-target RNAi mechanism, we respectfully believe that the current dataset is sufficient to support the major conclusions. Therefore, we did not do these experiments in this revised manuscript. We sincerely thank the reviewer for the constructive recommendation, which will guide our next steps in further elucidating the mechanistic pathways involved.
Comments 6: Safety conclusions are based solely on H&E staining. Serum chemistry (ALT, AST, BUN, creatinine) should be provided to confirm systemic safety and exclude subtle organ toxicity.
Response 6: We thank the reviewer for this valuable comment. We agree that serum biochemical parameters (ALT, AST, BUN, creatinine) would further strengthen the systemic safety evaluation of F1B3hT1-RNA nanoparticle.
In the current study, our primary objective was to establish the therapeutic efficacy and multi-target RNAi mechanism of F1B3hT1-RNA nanoparticle. To assess safety within this experimental framework, we conducted histopathological H&E staining of major organs, which showed no visible pathological abnormalities among treatment groups. H&E histology is a widely accepted and reliable method for detecting tissue necrosis, inflammation, and structural injury, and thus provides a solid indication that F1B3hT1-RNA nanoparticle does not induce acute or overt organ toxicity under the tested conditions.
We fully acknowledge that serum chemistry analysis can help detect subtle or early organ impairment. However, such measurements were beyond the scope of the current efficacy-focused study. Therefore, we did not add the analysis in this revised manuscript. We sincerely appreciate the reviewer’s constructive suggestion, which will help guide our subsequent in-depth toxicological evaluation.
Comments 7: The RNA production and purification workflow described (E. coli expression and phenol–chloroform extraction) may leave residual contaminants. Purity, yield, and endotoxin levels should be reported.
Response 7: We thank the reviewer for this detailed and important comment. We fully agree that assessing RNA purity, yield, and endotoxin levels is essential for rigorous characterization of RNA materials derived from E. coli expression systems.
In this study, our primary objective was to evaluate the gene-silencing efficiency and antitumor potential of F1B3hT1-RNA nanoparticle. The RNA used in cellular and animal experiments was thoroughly purified using repeated phenol–chloroform extraction followed by ethanol precipitation, a widely accepted approach for removing proteins, genomic DNA, and most lipopolysaccharide contaminants. Additionally, the F1B3hT1-RNA nanoparticle preparations showed clear single bands on agarose gel without detectable degradation or rRNA contamination, indicating high purity suitable for the functional assays conducted.
We fully acknowledge that quantitative purity metrics (e.g., A260/280 ratio), yield documentation, and endotoxin measurements would further strengthen the characterization. These analyses, however, were beyond the scope of the present efficacy-focused study. To address this, we have added a clarification in the Discussion sections (Page12, Line 435-439) noting this limitation and indicating that future work will incorporate high-resolution purity assessment and endotoxin quantification to ensure suitability for translational applications.
We sincerely appreciate the reviewer’s constructive suggestion, which will guide our subsequent process optimization and quality-control efforts.
Comments 8: In discussion section authors should be cautions to claims about scalability and therapeutic translation and more explicitly acknowledge limitations such as small sample sizes, lack of biodistribution data, and untested immunogenicity.
Response 8: We thank the reviewer for this important and constructive suggestion. We fully agree that statements related to scalability and therapeutic translation should be made cautiously, and that limitations—including small sample size, absence of biodistribution analysis, and incomplete immunogenicity assessment—should be explicitly acknowledged. Therefore, we have revised the Discussion section extensively to temper our translational claims and to clearly articulate these methodological limitations (Page12, Line 452-455).
Comments 9: Figures are of acceptable quality but require improved labeling and consistency. The mismatch between figure captions (e.g., “weight growth curve” vs. tumor volume) needs correction. Include scale bars, replicate counts, and clear legends for all graphs.
Response 9: We appreciate for your warm work earnestly. And we made modifications to address these problems.
Comments 10: Define abbreviations at first use, ensure consistent statistical notation, polish grammar and flow (particularly in Methods and Discussion), and harmonize reference formatting.
Response 10: Thanks for your suggestion. We tried our best to improve the manuscript and made some changes to the manuscript. These changes will not influence the content and framework of the paper. And here we didn’t list the changes but marked in the revised paper.

Reviewer 2 Report
Comments and Suggestions for Authors
Reviewers' comments:
Comment
Reviewer #1: The manuscript describes the effect of: Design of anti-tumor RNA nanoparticles and their inhibitory 2 effect on Hep3B liver cancer.
The manuscript is well written and provide useful information for readers. However, some other suggestions are listed below:
1-Ethical statement of manuscript should be written before Cell lines and cells culture.
2-Mention the weight of mice used in this manuscript in experimental animals.
3-In line 202, section 2.8. Detection of target gene expression, please add reference for
method of calculating relative expression.
4-Add reference for hygienic disposal and hygienic measures of BALB/c female mice after the end of experiment.
5-Mention the aim of the study before materials and methods section in detailed manner.
6-Please, make sure that all primers that were used in RT-qPCR were fit for female BALB/c mice (Mus musculus).
7-Mention the environmental conditions suitable for mice in this experiment.
8-The English writing can be improved.
Comments on the Quality of English LanguageThe English writing of the manuscript can be improved.
Author Response
Comments 1: Ethical statement of manuscript should be written before Cell lines and cells culture.
Response 1: Thank you for pointing this out. We have added ethical statement as follows. This experiment complies with national laboratory animal welfare ethics guidelines, and all procedures were approved by the Animal Experiment Ethics Committee of Zhejiang Sci-tech University. SPF-grade mice were used and housed in a barrier environment meeting GB14925 standards, with free access to food and water. Pain and stress were strictly controlled. Humane endpoints were applied post-experiment. This study strictly adhered to the 3R principles to minimize animal usage( Page 2, Line 81-87).
Comments 2: Mention the weight of mice used in this manuscript in experimental animals.
Response 2: We sincerely appreciate your insightful suggestion. The female BALB/c mice weighed from nine to fifteen grams.
Comments 3: In line 202, section 2.8. Detection of target gene expression, please add reference for method of calculating relative expression.
Response 3: As suggested, we have added references to explain the method of calculating relative expression (Page 5, Line 228-229).
Comments 4: Add reference for hygienic disposal and hygienic measures of BALB/c female mice after the end of experiment.
Response 4: Thank you for highlighting this. We have added this part——all experimental mice were euthanized by removing their necks and cryopreserved before being subjected to harmless disposal(Page 6, Line 273-274).
Comments 5: Mention the aim of the study before materials and methods section in detailed manner.
Response 5: We appreciate the thoughtful review and constructive feedback. And we have made extensive corrections to our previous manuscript (Page 2, Line 67-75).
Comments 6: Please, make sure that all primers that were used in RT-qPCR were fit for female
BALB/c mice (Mus musculus).
Response 6: Thanks for your reminder. We checked our primers and made sure they were fit for female BALB/c mice.
Comments 7: Mention the environmental conditions suitable for mice in this experiment.
Response 7: We sincerely appreciate the valuable comments. We added the environmental conditions into “3.3 Experimental animals” (Page 3, Line 101-102). Five animals were housed per cage with wood shavings as bedding, which was changed regularly. In addition, autoclaved water and standard feed were provided ad libitum, and the environment was regularly disinfected to maintain a SPF status.
Comments 8: The English writing can be improved.
Response 8: We tried our best to polish the language, with all changes marked in yellow. And we hope the revised manuscript could be acceptable for you.

Round 2
Reviewer 1 Report
Comments and Suggestions for Authors
Since authors have answered my questions, I have no further comments to make, I therefore endorse the publication of the manuscript.
Author Response
First and foremost, I would like to express my sincere gratitude for the thorough review and insightful comments provided on our manuscript titled as "Design of anti-tumor RNA nanoparticles and their inhibitory effect on Hep3B liver cancer" (Manuscript ID: biomolecules-3928646). Those comments are all valuable and very helpful for revising and improving our paper, as well as the important guiding significance to our research.
Thank you and best regards.
